# Transcriptome-wide RNA binding analysis of C9orf72 poly(PR) dipeptides

Rubika Balendra[1,2,*], Igor Ruiz de los Mozos[3,4,5,*], Hana M Odeh[6], Idoia Glaria[1,2,7], Carmelo Milioto[1,2], Katherine M Wilson[1,2], Agnieszka M Ule[5], Martina Hallegger[3], Laura Masino[8], Stephen Martin[8], Rickie Patani[3,5], James Shorter[6], Jernej Ule[3,5,9], Adrian M Isaacs[1,2]

**An intronic GGGGCC repeat expansion in _C9orf72_ is a common genetic cause of amyotrophic lateral sclerosis and frontotemporal dementia. The repeats are transcribed in both sense and antisense directions to generate distinct dipeptide repeat proteins, of which poly(GA), poly(GR), and poly(PR) have been implicated in contributing to neurodegeneration. Poly(PR) binding to RNA may contribute to toxicity, but analysis of poly(PR)-RNA binding on a transcriptome-wide scale has not yet been carried out. We therefore performed crosslinking and immunoprecipitation (CLIP) analysis in human cells to identify the RNA binding sites of poly(PR). We found that poly(PR) binds to nearly 600 RNAs, with the sequence GAAGA enriched at the binding sites. In vitro experiments showed that poly(GAAGA) RNA binds poly(PR) with higher affinity than control RNA and induces the phase separation of poly(PR) into condensates. These data indicate that poly(PR) preferentially binds to poly(GAAGA)-containing RNAs, which may have physiological consequences.**

## Introduction

A hexanucleotide repeat expansion in the _C9orf72_ gene is the most common genetic cause of amyotrophic lateral sclerosis (ALS) and frontotemporal dementia (FTD) (DeJesus-Hernandez et al, 2011; Renton et al, 2011). Several mechanisms of toxicity have been implicated in contributing to the disease process (Balendra & Isaacs, 2018). Dipeptide repeat proteins (DPRs) produced by repeat-associated non-ATG (RAN) translation are likely to represent an important toxic entity (Ash et al, 2013; Mori et al, 2013; Zu et al, 2013). Five different DPRs are produced: poly(GA), poly(GP), poly(GR), poly(PA), and poly(PR). Of these, the arginine-containing DPRs, poly(GR) and poly(PR), are the most toxic in model systems

(Moens et al, 2017). The common pathological hallmark identified in the vast majority of sporadic and genetic ALS cases and a large proportion of FTD cases is mislocalisation and aggregation of the RNA- and DNA-binding protein TDP-43. This pathology is also found in _C9orf72_ ALS and FTD (C9FTD/ALS) and is likely to be downstream of DPR pathology (Balendra & Isaacs, 2018).

Several mechanisms have been attributed to DPR pathology and include nucleocytoplasmic trafficking dysfunction, DNA damage, and translational inhibition. A number of studies have explored the effect of the arginine-containing DPRs on membraneless organelles, such as stress granules and nucleoli. Interactome studies have confirmed that poly(PR) and poly(GR) bind to proteins enriched in prion-like low-complexity domains (LCDs), many of which are RNA-binding proteins (RBPs) and constituents of membraneless organelles (Lee et al, 2016; Lin et al, 2016; Boeynaems et al, 2017; Hartmann et al, 2018; Moens et al, 2019; Odeh & Shorter, 2020). LCDs in RBPs facilitate the process known as phase separation, by which membraneless organelles are formed, and this process is promoted by the presence of RNA (Molliex et al, 2015; Murakami et al, 2015; Patel et al, 2015; Protter et al, 2018). Mutations in TDP-43, FUS, and hnRNPA1 cause ALS/FTD, and these mutations are often localised within the LCDs of these RBPs. These mutations increase the formation of amyloid-like fibrils and disturb phase separation dynamics. Poly(PR) and poly(GR) disrupt the dynamics of phase separation in membraneless organelles in cells and impair translation (Lee et al, 2016; Boeynaems et al, 2017; Hartmann et al, 2018; Zhang et al, 2018; Moens et al, 2019; White et al, 2019). These arginine-rich DPRs can also undergo phase separation themselves in vitro, which is dependent on anion charge, and the presence of RNA dose-dependently increases the phase separation of poly(PR) (Boeynaems et al, 2017; Boeynaems et al, 2019). It is possible that these interactions with RBPs and other LCD-containing proteins are partly mediated by interactions of poly(PR) with

[1]UK Dementia Research Institute at UCL, London, UK   [2]Department of Neurodegenerative Disease, UCL Queen Square Institute of Neurology, London, UK   [3]The Francis Crick Institute, London, UK   [4]Department of Personalized Medicine, NASERTIC, Government of Navarra, Pamplona, Spain   [5]Department of Neuromuscular Diseases, UCL Queen Square Institute of Neurology, London, UK   [6]Department of Biochemistry and Biophysics, Perelman School of Medicine at the University of Pennsylvania, Philadelphia, PA, USA   [7]Research Support Service, Institute of Agrobiotechnology, CSIC-Government of Navarra, Mutilva, Spain   [8]Structural Biology Science Technology Platform, The Francis Crick Institute, London, UK   [9]UK Dementia Research Institute at King's College London, Maurice Wohl Clinical Neuroscience Institute, London, UK

Correspondence: a.isaacs@ucl.ac.uk; jernej.ule@kcl.ac.uk
*Rubika Balendra and Igor Ruiz de los Mozos contributed equally to this work

RNA. An interactome analysis of poly(GR)80 expressed in human embryonic kidney cells revealed that it interacts with RBPs and ribosomal proteins, including mitochondrial ribosomal proteins (Lopez-Gonzalez et al, 2016). Several interactions were abolished when samples were treated with RNase A, suggesting some were RNA-mediated. Another study demonstrated that poly(PR) interacts with multiple DEAD-box RNA helicases, and that this is dependent on RNA, suggesting RNA mediates the interaction (Suzuki et al, 2018). Poly(PR)20 peptide, when applied exogenously to human astrocyte cells in culture, leads to alterations in splicing of several mRNAs and a change in abundance of mRNAs encoding ribosomal proteins in particular (Kwon et al, 2014), and some of these RNAs are bound directly by poly(PR) (Kanekura et al, 2016).

However, a transcriptome-wide analysis of poly(PR) binding to RNAs in the cellular context has not been investigated yet. To achieve this goal, we used improved iCLIP (iiCLIP), which enables quantitative identification of protein–RNA crosslinking sites in vivo (Lee et al, 2021 *Preprint*), to investigate whether the arginine-containing DPR poly(PR) binds to RNA with some sequence specificity in human cells. We show that poly(PR) directly crosslinks to RNA and shows enriched crosslinking on specific transcripts, including ALS-relevant mRNAs such as neurofilament medium chain (*NEFM*) and nucleolin (*NCL*). We further show that poly(PR) interacts with nanomolar affinity with GAAGA-containing RNA, which also promotes the phase separation of poly(PR).

## Results

### Poly(PR) iiCLIP reveals binding to specific RNAs

To investigate transcriptome-wide DPR binding to RNA, we established denaturing purification of DPR-RNA complexes for CLIP based on the previously established approach (Fig 1A–D) (Huppertz et al, 2014; Lee et al, 2021 *Preprint*). We expressed doxycycline-inducible triple FLAG-tagged PR100 or GA100, or triple FLAG tag alone in human embryonic kidney cells (HEK293Ts) (Figs 1B and S1). PR100-FLAG and GA100-FLAG were both present in the nucleus and cytoplasm, with PR100-FLAG having greater nuclear localisation than GA100-FLAG (Fig S1A and B). There was no difference in transfection efficiency or expression level between PR100-FLAG and GA100-FLAG (Figs 1C and S1C), and FLAG was detected by a dot blot in FLAG-expressing cells (Fig S1D–F). After 24 h of exogenous expression, we used UV light to crosslink protein–RNA interactions. To test for specific binding of poly(PR) to RNA, we used the control conditions of poly(PR)-expressing cells, which were non-crosslinked, and FLAG-expressing cells, which were crosslinked. We subsequently immunoprecipitated the DPR-RNA complexes using the FLAG tag (Fig S2A and B). We then employed the iiCLIP protocol to ligate an infrared adaptor for visualisation of the protein–RNA complexes, and extracted and reverse-transcribed the RNA to generate cDNA libraries for high-throughput sequencing (Lee et al, 2021 *Preprint*). Infrared visualisation of the DPR-RNA complexes showed much stronger signal in the

crosslinked PR100-FLAG cells (Fig 2A, lanes 3 and 4) compared with the non-crosslinked PR100-FLAG cells (Fig 2A, lane 5), the crosslinked GA100-FLAG cells (Fig 2A, lanes 6 and 7), and the crosslinked FLAG-only cells (Fig 2A, lane 9)—and this difference is especially apparent for the diffuse signal that usually represents proteins crosslinked to longer RNA fragments (Fig 2A). This finding indicates that PR100 directly crosslinks to RNA in human cells.

Sequencing of the iiCLIP reads revealed over 1,200,000 unique cDNA crosslinking events in the crosslinked PR100-FLAG cells across multiple replicates, with significantly fewer in the control (<74,000) and GA100 (<120,000) conditions (Fig 2B). PR100 crosslinking events occurred most frequently in introns, intergenic regions, and the coding sequence, with additional signal in noncoding RNAs and 3′ and 5′ UTRs (Fig 2C). We identified 558 mRNAs with high levels of binding (≥200 crosslinking events) to PR100 as compared to the controls of PR100-non-crosslinked and FLAG-alone samples (Table S1). Examples of genes with the highest numbers of binding events (>1,000 crosslinking events) included X-inactive specific transcript (*XIST*), metastasis-related lung adenocarcinoma transcript 1 (*MALAT1*), *NEFM*, *NCL*, nuclear enriched abundant transcript 1 (*NEAT1*), and heterogeneous nuclear ribonucleoprotein U (*HNRNPU*) (Table S1). *XIST* and *NEAT1* were in the top seven genes with the largest number of crosslinking events (Table S1), in agreement with their previously identified interaction with poly(PR) through RNA-IP experiments (Suzuki et al, 2019). In addition, the parkin gene (*PRKN*), mutations in which cause Parkinson's disease (Kitada et al, 1998), had 266 crosslinking events. Gene Ontology (GO) enrichment analysis of the 558 mRNAs with the highest number of crosslinks to PR100 (Table S1) revealed significant enrichment involving the biological processes of "RNA splicing," "regulation of chromosome organisation," and "covalent chromatin modification" (Fig 2D and E). There was also significant enrichment in the cellular components of "nuclear speckles," "chromosome region," and "centromeric chromosome region" (Fig S3A and B), and in the molecular functions of "ATPase activity," "DNA-dependent ATPase activity," and "helicase activity" (Fig S3C).

### Poly(PR) binds with high affinity to poly(GAAGA) RNAs

We analysed enrichment of 5-mer motifs in our PR100 iiCLIP dataset, which identified GAAGA as a highly enriched pentameric sequence (Fig 3A), exemplified in the *NCL* and *NEFM* transcripts (Fig 3C and D). This enrichment appeared specific for PR100, as enrichment of 5-mer motifs in the GA100 iiCLIP dataset identified CCGGG as the most enriched pentamer (Fig S4). AUAAU was a less-represented motif (in the bottom 5% of all motifs) surrounding the PR100 crosslinking site (Fig 3B). To determine whether there was a differential affinity of poly(PR) for these 5-mer RNA sequences, we used biolayer interferometry to compare the affinity of purified PR20 and GP20 peptides with biotinylated RNA oligonucleotides containing five repeats of GAAGA or AUAAU (Table 1). PR20 had a stronger affinity for the poly(GAAGA) RNA with an apparent $K_d$ of 2.6 ± 0.5 nM (Fig 4A, E, and I) compared with the poly(AUAAU) RNA, with an apparent $K_d$ of 11.1 ± 2.5 nM (Fig 4C and G). There was no evidence of interaction between the DPR GP20 and the poly(GAAGA) RNA, even at 180-fold higher concentrations (3–25 $\mu$M

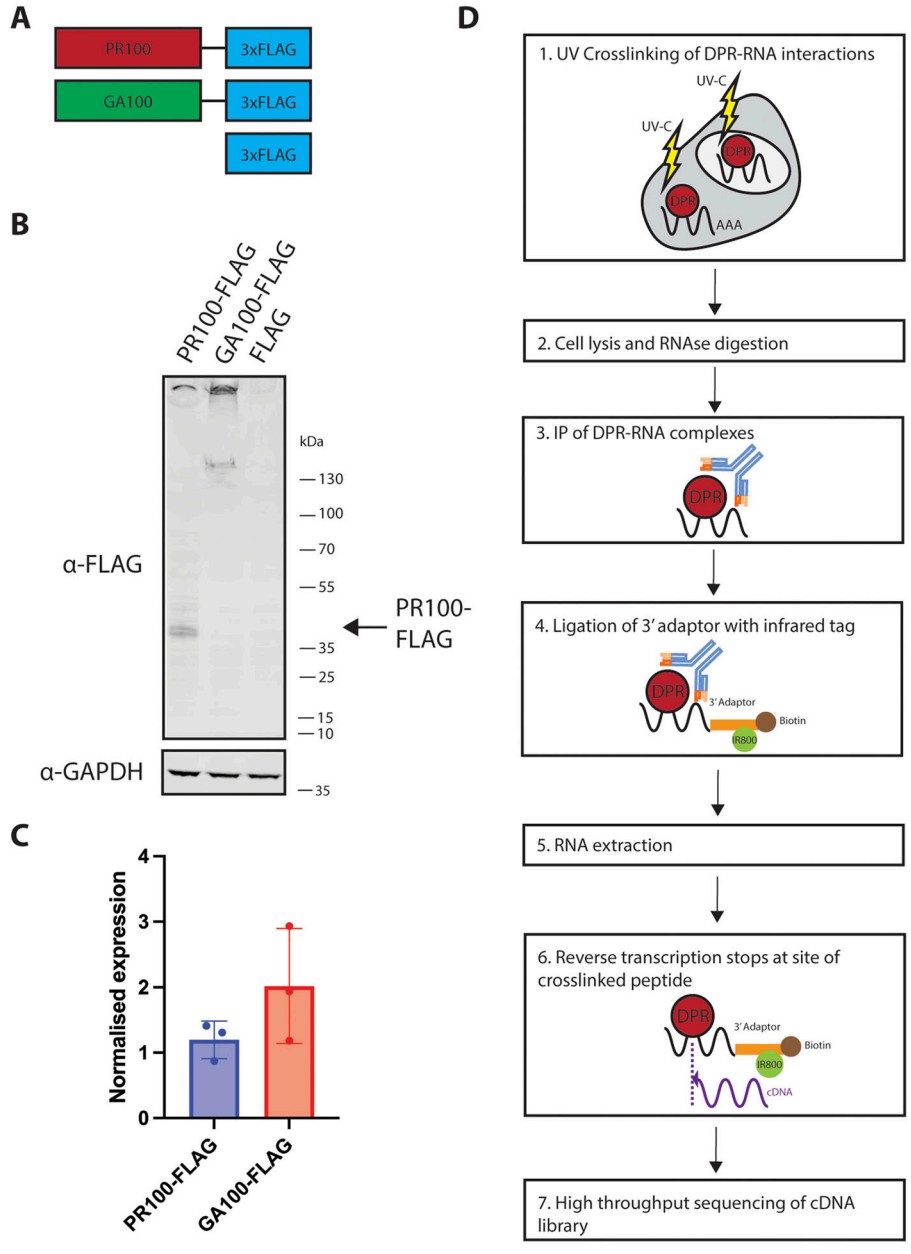

**Figure 1. C9orf72 dipeptide repeat proteins (DPRs) iiCLIP pipeline.**
**(A)** Diagram of the PR100-3xFLAG, GA100-3xFLAG, and 3xFLAG-alone constructs used in this study. **(B)** Anti-FLAG immunoblot 24 h post-induction of PR100-3xFLAG, GA100-3xFLAG, and 3xFLAG in HEK293T cells. Note GA100 appears mostly aggregated as the majority is present at the top of the gel and FLAG alone is not visible because of its low molecular weight, so its expression was confirmed by a dot blot (Fig S1D–F). **(C)** Quantification of the expression of PR100-FLAG and GA100-FLAG normalised to GAPDH expression. No difference was observed between these conditions. n = 3 replicates per condition. Bars show the average and SD. $P > 0.05$, two-tailed unpaired $t$ test. **(D)** Summary of the iiCLIP pipeline for investigation of DPR-RNA direct interaction. Transiently transfected HEK293Ts were UV-crosslinked to stabilise DPR–protein interactions, and cells were lysed and digested with RNase. The FLAG tag was used for immunoprecipitation of DPR-RNA complexes. A preadenylated, infrared dye–labelled adaptor was ligated onto the 3' end of the RNA. RNA was extracted and reverse-transcribed, generating cDNA libraries, which were high-throughput-sequenced, and the data were analysed to determine sites of binding with nucleotide specificity across the transcriptome.

for GP20 compared with up to 133 nM for PR20) (Fig 4K). We also examined the affinity of purified GR20 peptides for the poly(GAAGA) or poly(AUAAU) RNA. GR20 had a slightly stronger affinity for the poly(GAAGA) RNA with an apparent $K_d$ of 3.6 ± 0.9 nM (Fig 4B, F, and J) compared with the poly(AUAAU) RNA, with an apparent $K_d$ of 6.1 ± 0.6 nM (Fig 4D and H). The difference in affinities between GAAGA and the control RNA sequence was higher for poly(PR) than for poly(GR) (Fig 4I and J), which is consistent with the pentamers being derived from poly(PR) iiCLIP data. These experiments show that both poly(PR) and poly(GR) have a high binding affinity for the tested RNAs, as expected because of their positive charge. Interestingly, this binding shows some sequence specificity, as a higher affinity was observed with the GAAGA motif that was most enriched in the iiCLIP experiment.

### Poly(GAAGA) RNA enhances poly(PR) and poly(GR) phase separation

Because of the high affinity of the poly(GAAGA) RNA sequence for poly(PR) and poly(GR), and its enrichment in poly(PR) binding sites in vivo, we investigated whether the poly(GAAGA) RNA could influence poly(PR) and poly(GR) phase separation. Poly(PR) and poly(GR) undergo phase separation in the presence of polyanions, such as RNA (Boeynaems et al, 2017; Boeynaems et al, 2019; Hutten

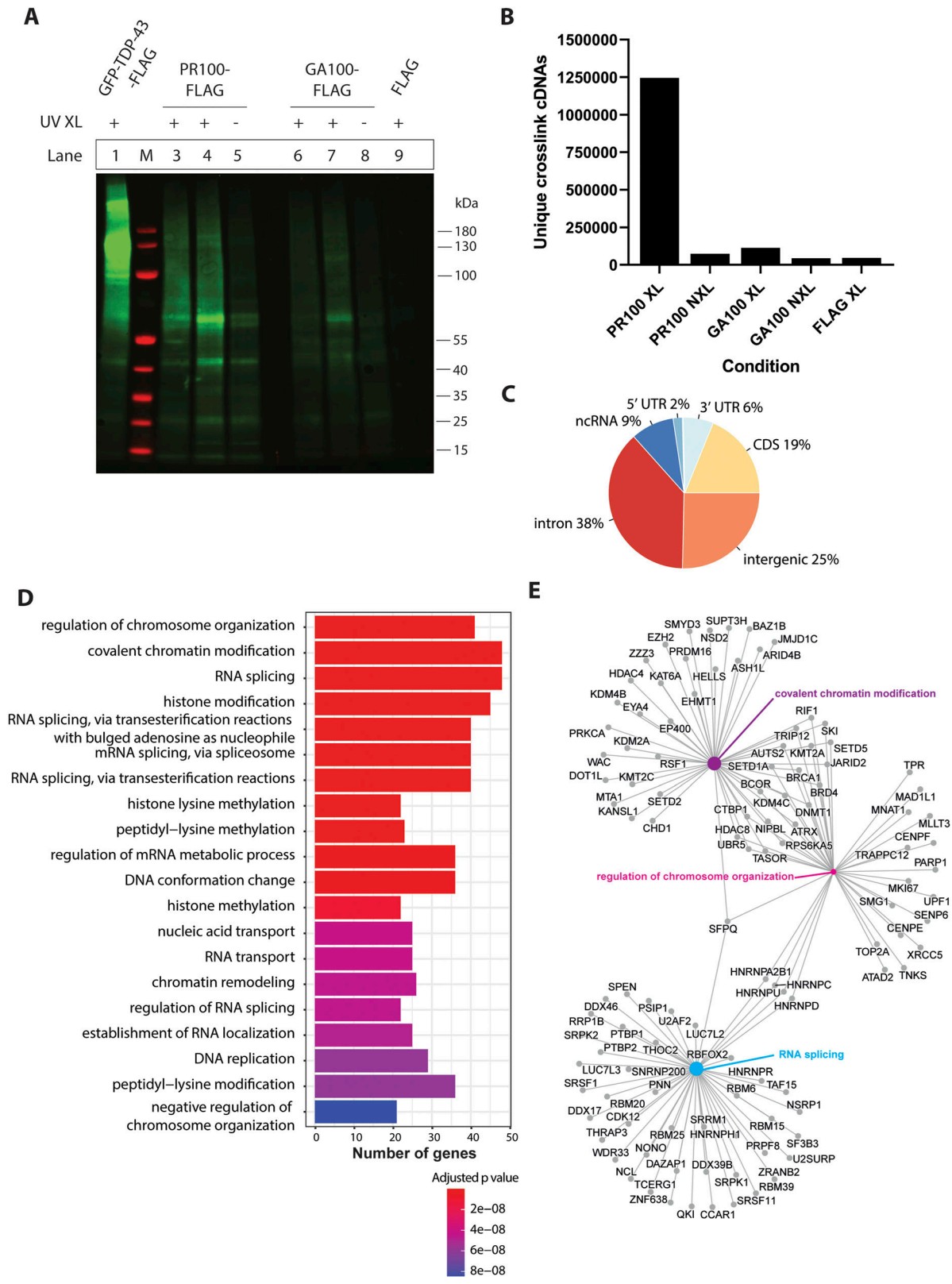

**Figure 2. iiCLIP reveals poly(PR) binds to RNA in human cells.**
**(A)** Infrared labelled protein–RNA complexes were separated by SDS–PAGE and transferred onto a nitrocellulose membrane. In lane 1, GFP-TDP-43-FLAG was run as a positive control, with a diffuse smear detected above its molecular weight, representing the GFP-TDP-43-FLAG-RNA complexes. Lane M is the protein ladder marker. Lanes 3 and 4 are crosslinked PR100-FLAG cells, which have a higher intensity than the PR100-FLAG–non-crosslinked cells (lane 5), the GA100-FLAG–crosslinked cells

et al, 2020). Thus, we examined whether poly(GAAGA) RNA had an effect on poly(PR) and poly(GR) phase separation. In the absence of RNA, poly(PR) and poly(GR) do not undergo phase separation (Fig 5A, left). Remarkably, the presence of equimolar poly(GAAGA) significantly increased poly(PR) and poly(GR) phase separation, indicated by a marked increase in turbidity, and numerous, small round, translucent condensates (Fig 5A, middle, and Fig 5B). In contrast, the poly(AUAAU) RNA induced fewer and larger poly(PR) and poly(GR) condensates, which were morphologically distinct from those formed in the presence of poly(GAAGA) RNA (Fig 5A, right). However, poly(AUAAU) RNA induced less phase separation than poly(GAAGA) RNA for both poly(PR) and poly(GR) (Fig 5B). These findings suggest that higher affinity RNA, such as poly(GAAGA), displays enhanced ability to induce poly(PR) and poly(GR) condensation.

# Discussion

In this study, we have investigated whether poly(PR) produced in C9FTD/ALS may directly bind to RNA in human cells. Indeed, we demonstrated direct and specific interactions between the arginine-rich DPR poly(PR) and GAAGA-containing RNAs using a transcriptome-wide approach. Arginine-rich DPRs have been shown to exert deleterious effects on several cellular functions, which include nucleocytoplasmic transport (Freibaum et al, 2015; Jovicic et al, 2015; Zhang et al, 2015; Boeynaems et al, 2016), phase transition of cellular organelles (Lee et al, 2016; Lin et al, 2016; Boeynaems et al, 2017), proteostasis (Kramer et al, 2018), and RNA dysregulation, which has previously been described in C9FTD/ALS models and patient cells and tissues (Kwon et al, 2014; Kanekura et al, 2016; Yin et al, 2017). Some of these effects are likely to be caused by the interactions of arginine-rich DPRs with other proteins, and our study suggests that their RNA interactions might also contribute to these effects.

Using in vitro studies, we confirmed that poly(PR) binds RNA with nanomolar affinity, with a stronger apparent affinity for the poly(GAAGA) as compared to poly(AUAAU) RNAs. Further to the discovery that poly(U) RNA promotes the phase separation of poly(PR) (Boeynaems et al, 2017), it has been shown that in a test tube, poly-rA, poly-rU, and poly-rC RNA homopolymers can promote the phase separation of poly(PR), but poly-rG does so to a lesser extent (Boeynaems et al, 2019). In the homopolymeric form, the affinity of the interaction between poly(PR) and poly-rA is the strongest, whereas poly(PR) has an almost identical affinity for poly-rU and poly-rC and the lowest affinity for poly-rG. This finding has been explained by the ability of poly-rG to form G-quadruplex structures, as opposed to other homopolymeric RNAs, which are unstructured. It has been hypothesised that base stacking interactions associated with G-quadruplex formation could compete with the poly(PR)

interaction. Intriguingly, in comparison with these previous findings, the pentameric sequence we found to be the most enriched binding to poly(PR) transcriptome-wide in vivo has a higher G-content (but with insufficient guanines to form G-quadruplexes) than the least frequently bound pentamer, suggesting that RNA sequences containing guanine can have a high affinity for poly(PR) in the cellular context. Furthermore, it was demonstrated that mixing homopolymeric RNA molecules, which can make complementary base pairs, can change the interactions between RNA and poly(PR), possibly because of competition between RNA base pairing interactions and RNA–peptide interactions (Boeynaems et al, 2019). Of importance, adding total HEK cell RNA dose-dependently ameliorates a nuclear import phenotype induced by adding poly(GR) and poly(PR) to cells (Hayes et al, 2020), suggesting RNA may reduce these phenotypes through high-affinity interactions with these DPRs. Intriguingly, we found that RNA sequences that tightly bind to poly(PR) with high affinity have an increased ability to promote poly(PR) condensate formation. It would be of interest in future studies to determine whether these RNA-induced condensates are less toxic to cells and whether poly(GAAGA) is able to alter the phase separation of poly(PR) within cells. One appealing possibility would be to use PR-specific RNA sequences, such as poly(GAAGA), as "baits" to safeguard the cell by sequestering poly(PR) from deleterious interactions. In fact, TNPO1, a nuclear import receptor, has been shown to play such a protective role against DPRs when overexpressed (Hutten et al, 2020).

Although it was our intention to provide a transcriptome-wide dataset rather than to focus on specific transcripts bound by poly(PR), we report that several interesting RNAs are bound. These include the previously identified paraspeckle long non-coding RNA NEAT1, for which poly(PR) binding was shown to lead to NEAT1 up-regulation (Suzuki et al, 2019), and NCL, which encodes the nucleolin protein, which is known to have a more dispersed nuclear localisation in C9orf72 human tissue and disease models (Haeusler et al, 2014). As expected, based on previous studies (Kwon et al, 2014), PR showed some punctate nuclear staining consistent with nucleolar localisation and this is likely to greatly influence the RNAs that are found to be bound to it. This study has also been performed in human cell lines using the overexpression of poly(PR), which limits the disease relevance of these findings, and future studies in neuronal models with more physiological expression levels would be of importance. It is increasingly recognised that RNA dysregulation plays a major role in ALS/FTD and many genetic causes of ALS/FTD are in RBPs (Nussbacher et al, 2019). Our dataset of poly(PR)–RNA binding can now be used for hypothesis-driven investigation of poly(PR) effects on RNAs. Understanding the biology of these interactions may help to further elucidate the underlying aetiology of neurodegeneration in C9FTD/ALS.

---

(lanes 6 and 7), the GA100-FLAG–non-crosslinked cells (lane 8), and the FLAG-crosslinked cells (lane 9). The FLAG tag consists of 3XFLAG. **(B)** Frequency of unique cDNAs, which represent individual crosslinking events identified by iiCLIP analysis. **(C)** Genomic location of PR100 binding sites, in introns, intergenic regions, and the coding sequence, with additional signal in non-coding RNAs and 5′ UTR and 3′ UTR segments. **(D)** Gene Ontology gene set enrichment analysis of PR100-crosslinked RNAs. Genes of RNAs bound in PR100 samples are represented by their Biological Process. The number of genes from the PR100-FLAG crosslinking dataset in each Gene Ontology category is shown and colour-coded by an adjusted $P$-value. **(E)** Genes of RNAs bound in PR100 samples from the top three significant categories within Biological Process (RNA splicing, regulation of chromosome organisation, and covalent chromatin modification) are represented in a gene-concept network. The size of the circle for each Biological Process is proportional to the number of genes identified within that category.

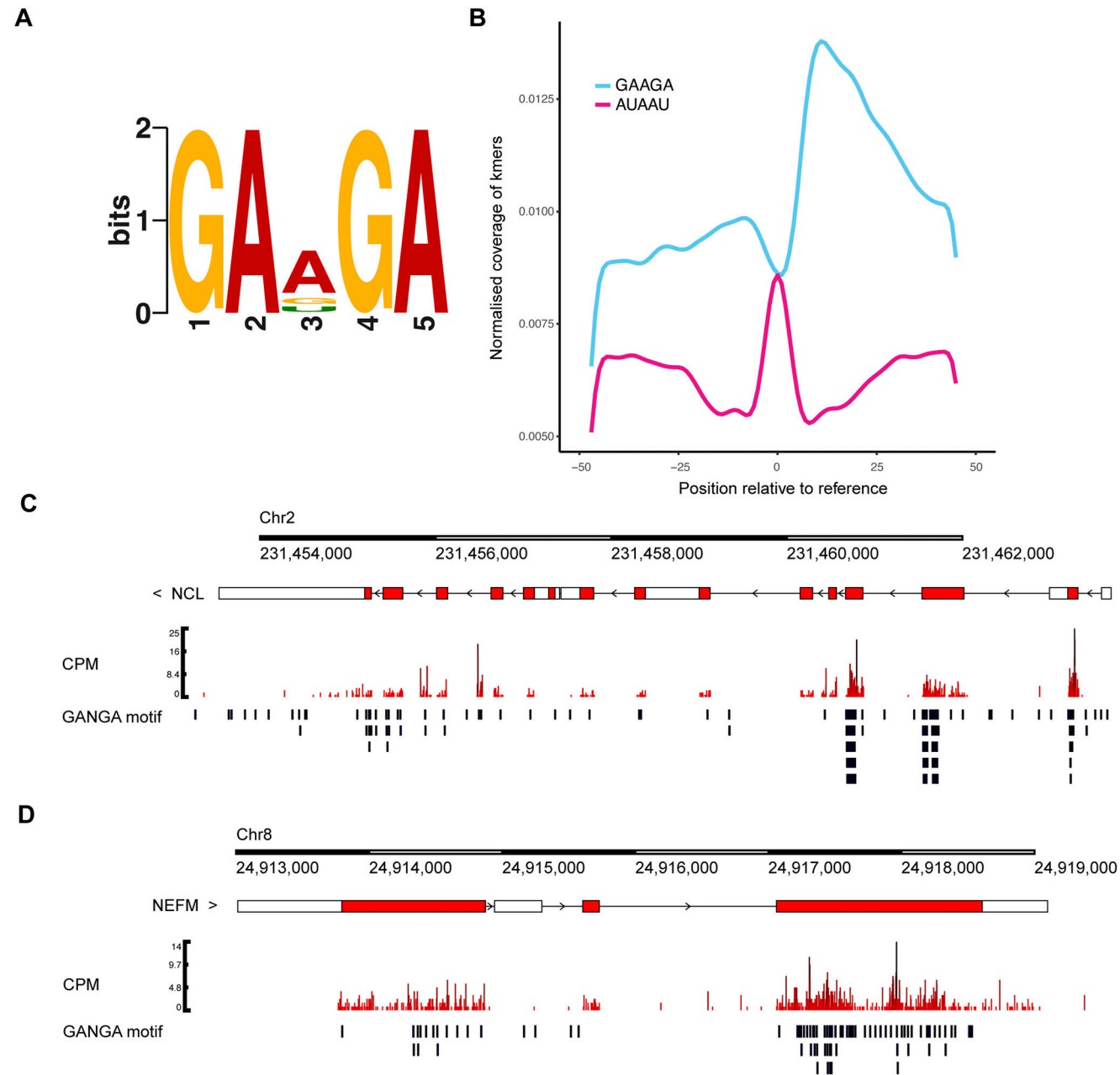

**Figure 3. Motif enrichment analysis of poly(PR)-RNA binding.**
**(A)** Motif enrichment analysis of the PR100 binding crosslinking sites revealed the most frequent pentamer bound by PR100 was GAAGA ($P$ = 3.1 × $e^{-930}$). **(B)** Analysis of the position of the pentamer relative to the crosslinking site. GAAGA is enriched upstream and downstream of the crosslinking site. The AUAAU pentamer has a lower frequency both upstream and downstream of the crosslinking site. **(C, D)** *NCL* and *NEFM*, which are transcripts highly bound by PR100 in the iiCLIP dataset (Table S1), have frequent GANGA (GAAGA, GAGGA, or GACGA) motifs in proximity to PR100 binding sites. The lower part of each panel indicates the position of these motifs relative to the crosslinking sites within the gene. Gene tracks were normalised by counts per million.

**Table 1. Oligonucleotide sequences.**

| | Biolayer interferometry RNA oligonucleotides |
|---|---|
| GAAGA sequence | 5'-/5Biosg/rGrArG rArArG rArGrA rArGrA rGrArA rGrArG rArArG rArGrA rA-3' |
| AUAAU sequence | 5'-/5Biosg/rArUrA rArUrA rUrArA rUrArU rArArU rArUrA rArUrA rUrArA rU-3' |

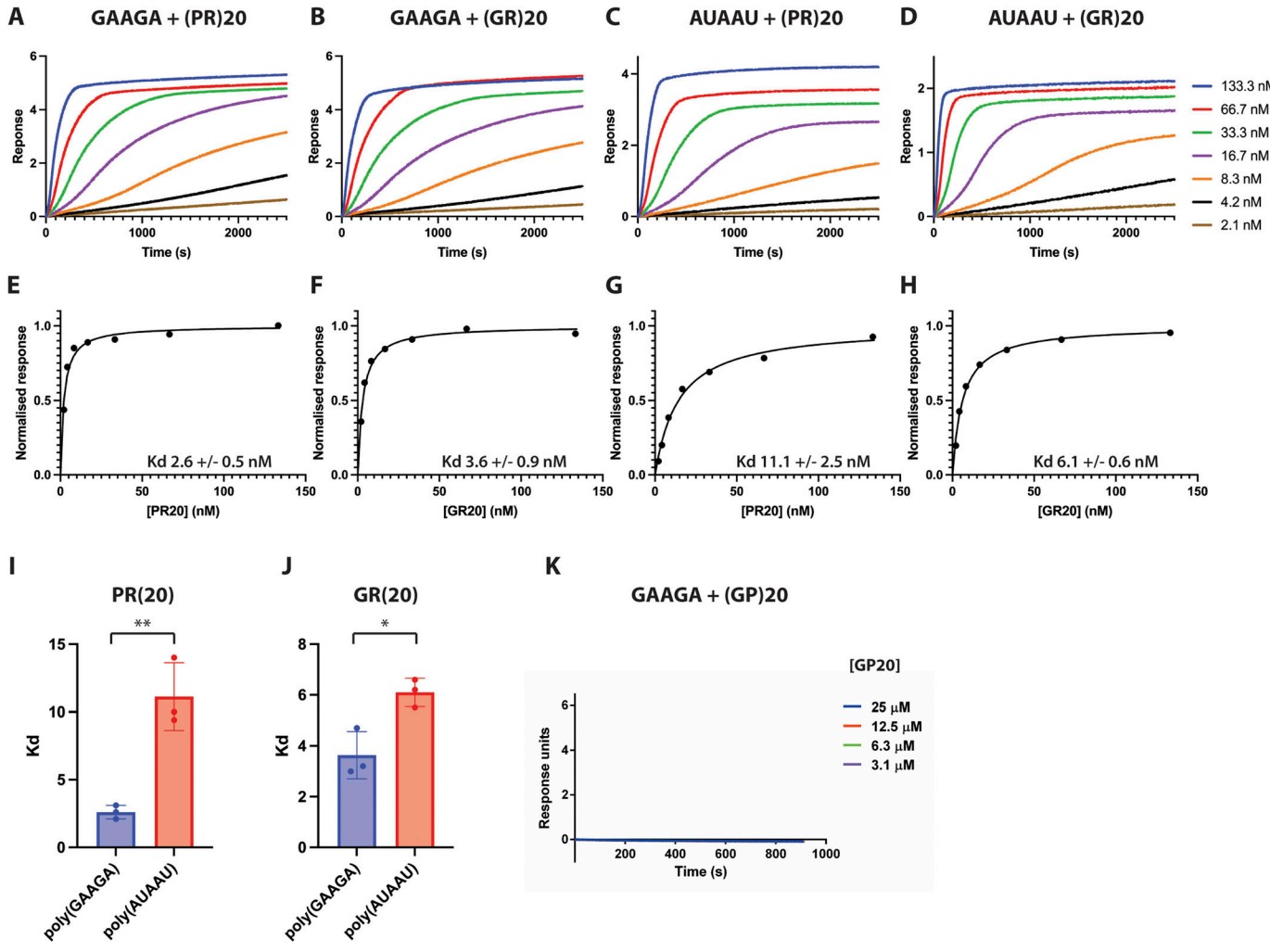

**Figure 4. Poly(PR) and poly(GR) directly bind to RNA with nanomolar affinity.**
Biolayer interferometry experiments measuring binding of RNA to PR20, GR20, and GP20. **(A, B, C, D)** Association phases of individual representative experiments. As dissociation was extremely slow (several hours), it was only partially recorded and it is not shown. **(E, F, G, H)** Plots of normalised response versus PR20 or GR20 concentration are shown. **(I, J)** $K_d$ of poly(GAAGA) and PR20-GR20 interaction was significantly higher than poly(AUAAU) and PR20-GR20 interaction. **(I, J)** Bars show the average and SD of $K_d$ of three independent replicate experiments for PR20 (I) and GR20 (J) with poly(GAAGA) or poly(AUAAU). **$P < 0.01$ and *$P < 0.05$, two-tailed unpaired $t$ test. **(K)** Biolayer interferometry experiments measuring binding of poly(GAAGA) to GP20. 180-fold higher concentrations for GP20 (3.1–25 $\mu M$) were used compared with PR20 and GR20 (2.1–133.3 nM) to confirm there was no interaction between GP20 and poly(GAAGA) RNA.

# Materials and Methods

### Cell lines

HEK293Ts were cultured in DMEM supplemented with 10% FBS, grown at 37°C with 5% $CO_2$, and routinely passaged.

### Transient transfections and iiCLIP protocol

PR100 and GA100 (Mizielinska et al, 2014) were cloned into pcDNA5 Flp-In Expression vectors with a 3′ triple FLAG tag, generating PR100-3xFLAG (PR100-FLAG) and GA100-3xFLAG (GA100-FLAG) pcDNA5 plasmids, with the 3xFLAG-only vector (FLAG) also used

as a control. For iiCLIP experiments, HEK293Ts were grown at ≈80% confluency in 10-cm plates and transiently transfected with PR100-FLAG, GA100-FLAG, or FLAG pcDNA5 plasmids using Lipofectamine 2000. The expression of the constructs was induced by supplementing the media with 150 ng ml⁻¹ of doxycycline for 24 h.

The iiCLIP protocol was performed as previously described (Lee et al, 2021 Preprint). Transiently transfected cells induced for 24 h were irradiated with UV once with 160 mJ/cm² using a Stratalinker 1800 at 254 nm. DNase was used after cell lysis to remove DNA. Protein–RNA complexes were ligated to a pre-adenylated, infrared dye–labelled adaptor and purified. RNA was isolated using proteinase K digestion and reverse-transcribed into cDNA. cDNA was subsequently purified and circularised.

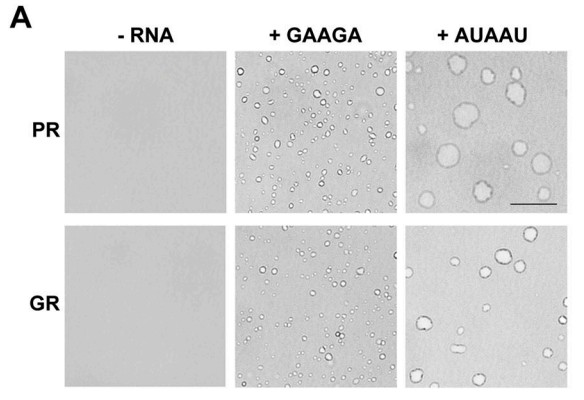

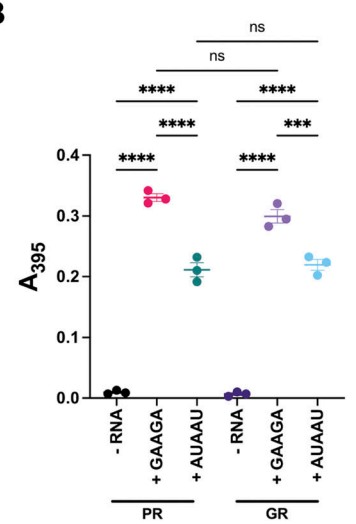

**Figure 5. Poly(GAAGA) RNA enhances poly(PR) and poly(GR) phase separation.** **(A)** Condensation of 20 μM PR20-FLAG or GR20-FLAG–alone (- RNA) or induction of equimolar concentrations of either poly(GAAGA) RNA or poly(AUAAU) RNA. Representative images from three independent experiments were taken using bright-field microscopy. The black bar represents 10 μm. **(B)** Turbidity (absorbance at 395 nm) measurements of 20 μM PR20-FLAG or GR20-FLAG with or without RNA. Values represent the mean of three independent experiments ± SEM. One-way ANOVA (***P = 0.0001; ****P < 0.0001; and ns, not significant).

### iiCLIP analysis

Multiplexed cDNA libraries were sequenced using Illumina HiSeq, generating 100-nt single-end reads. Sequenced reads were processed by the iMaps software package (http://icount.biolab.si/), and demultiplexed into individual libraries based on their experimental barcodes. Unique molecular identifier nucleotides were used to distinguish and collapse PCR duplicates. The barcode sequences and adaptors were removed from the 5′ and 3′ ends. Trimmed sequences were mapped to the human genome (build GRCh38, Gencode, version 27) with STAR aligner allowing two mismatches (Langmead & Salzberg, 2012). Uniquely mapping reads were kept, and the preceding aligned nucleotide was assigned as the DPR-crosslinked site. Significant crosslinking sites were determined by the iCount False Discovery Rate (<0.05) algorithm by weighing the enrichment of crosslinks versus shuffled random positions (https://github.com/tomazc/iCount). For subsequent analysis, we set a threshold of at least 300,000 unique crosslinking events for each PR100-crosslinked sample, and n = 4 samples met this threshold. For three of these samples, protein–RNA complexes had been purified using SDS–PAGE and transferred onto nitrocellulose membranes, as previously described (Lee et al, 2021 Preprint). For one of these samples, protein–RNA complexes had been purified using immunoprecipitation with beads. We analysed the number of crosslinking events for these PR100-crosslinked samples and their corresponding controls: PR100-non-crosslinked, GA100-crosslinked, GA100-non-crosslinked, and FLAG-crosslinked. Crosslinking events were normalised by the total number of crosslinks in the sample per million (counts per million). For gene-level analysis, genes were identified that contain at least 200 crosslinking events from PR100-FLAG iiCLIP, with <10% binding in the control conditions of PR100-FLAG–non-crosslinked samples and FLAG-crosslinked samples (Table S1). These genes were used for ontology enrichment analysis performed with the R package clusterProfiler, comparing against all other genes, using an FDR correction and adjusted P-value cut-off of <0.01 (Yu et al, 2012). Enriched pentamers

were calculated with DREME v.5.4.1 (Bailey, 2011) using the 5 nt upstream and 30 nt downstream of the significant crosslinking sites, compared with similar sequences collected from random positions of the same genes that did not overlap with any significant crosslinking sites. Pentamers were chosen as they have proven in previous systematic studies most useful to distinguish the sequence binding specificity of RBPs both for analysis of in vitro binding specificity from methods such as RNA Bind-n-Seq (Dominguez et al, 2018) and for analysis of CLIP data (Kuret et al, 2022), and thus, the results of our analyses can be most easily compared with other studies.

### Immunoblotting

Transfected HEK293Ts were homogenised in lysis buffer (50 mM Tris–HCl, pH 7.4, 100 mM NaCl, 1% Igepal CA-630, 0.1% SDS, and 0.5% sodium deoxycholate supplemented with cOmplete protease inhibitor cocktail [Roche]) and sonicated in a Bioruptor. Samples were centrifuged at 16,000g for 10 min. Supernatants were loaded with NuPAGE LDS sample buffer and DTT, then heated to 70°C for 5 min. Samples were separated on NuPAGE 4–12% Bis-Tris gels in MES running buffer, then transferred onto PVDF membranes. After blocking, membranes were incubated with anti-FLAG (F3165, 1:4,000; Sigma-Aldrich) or anti-GAPDH (2118S, 1:1,000; Cell Signaling Technologies) followed by complementary secondary antibodies (LI-COR IRDye, 1:10,000). Specific binding was detected with a LI-COR Odyssey CLx imager. The intensity of bands was quantified using Fiji–ImageJ software. Statistical analyses were performed using GraphPad Prism 9. Details are given in the figure legend. For dot blotting, supernatants of centrifuged samples prepared as above were dotted onto a nitrocellulose membrane (18 or 9 μg of total protein per dot). After blocking, membranes were incubated with anti-FLAG (F3165, 1:400; Sigma-Aldrich) or anti-GAPDH (2118S, 1:400; Cell Signaling Technologies) followed by complementary secondary antibodies (LI-COR IRDye, 1:10,000). Specific binding was detected with a LI-COR Odyssey CLx imager.

## Immunofluorescence staining

Transfected HEK293Ts grown on poly-D-lysine–coated PerkinElmer 96-well plates were fixed in 4% paraformaldehyde in PBS for 15 min, then washed three times in PBS supplemented with 0.3% Triton X (PBST), and blocked with 5% BSA in PBST for 1 h. Cells were incubated overnight at 4°C with a primary antibody anti-FLAG (F3165, 1:500; Sigma-Aldrich) in 5% BSA in PBST. Cells were washed in PBST three times, then incubated with a complementary Alexa Fluor secondary antibody (1:500) for 1 h at room temperature. Cells were washed once in PBST containing DAPI for 10 min, then twice more in PBST. Images were acquired using a Thermo Fisher Scientific CX5 high-throughput imaging microscope with a 10x objective. Images were analysed using proprietary onboard software. Statistical analyses were performed using GraphPad Prism 9. Details are given in the figure legend.

## Biolayer interferometry measurements

Biolayer interferometry experiments were performed on ForteBio Octet RED96 and Octet R8 instruments (Sartorius). Biotinylated RNAs were synthesised by Integrated DNA Technologies. PR20, GR20, and GP20 peptides were synthesised by Thermo Fisher Scientific. Biotinylated RNA and poly-DPR peptides were dissolved in Tris–EDTA (10 mM Tris–HCl and 1 mM disodium EDTA, pH 8.0) buffer solution with 150 mM NaCl, 0.1 mg/ml BSA, and 0.01% Tween-20 to reduce non-specific interactions. The assays were carried out at 25°C in a 96-well plate and a sample volume of 200 $\mu$l. Streptavidin-coated biosensors were pre-equilibrated, loaded with biotinylated RNAs, and exposed to protein concentrations ranging from 2.1 to 133 nM for PR20 and GR20 and from 3.1 to 25 $\mu$M for GP20. Equilibrium dissociation constants ($K_d$) for the RNA–protein interactions were determined by plotting the instrument response at equilibrium as a function of protein concentration and fitting the data assuming a 1:1 interaction, using non-linear least squares regression using Octet BLI Analysis software (Sartorius). Oligonucleotide sequences are provided in Table 1. Biological triplicates were performed using freshly prepared RNA and protein solutions in independent experiments. Statistical analyses were performed using GraphPad Prism 9. Details are given in the figure legend.

## In vitro poly(PR) and poly(GR) condensation assay

A poly(PR) 20-mer DPR with a C-terminal FLAG tag was purchased from CSBio and verified by mass spectrometry. The sequence of poly(PR) was as follows: PRPRPRPRPRPRPRPRPRPRPRPRPRPRPRPRPRPRPRPRPRGSFEG-DYKDDDDK. A poly(GR) 20-mer DPR with a C-terminal FLAG tag was purchased from DGpeptides. The sequence of poly(GR) was as follows: GRGRGRGRGRGRGRGRGRGRGRGRGRGRGRGRGRGRGRGRGRGSFEG-DYKDDDDK. Lyophilised powder was reconstituted in 1X PBS, and snap-frozen in single-use 200 $\mu$M aliquots and stored at –80°C. Poly(GAAGA) and poly(AUAAU) RNA sequences were ordered from IDT (sequences provided in Table 1). Lyophilised powder was reconstituted in RNase-free water, to a final stock concentration of 100 $\mu$M. Aliquots were snap-frozen and stored at –20°C. For RNA-induced condensate formation, poly(PR) or poly(GR) and all RNAs were first thawed on ice. Poly(PR) or poly(GR) was diluted to a final concentration of 20 $\mu$M in 20 mM Hepes–NaOH (pH 7.4), 150 mM NaCl, and 1mM DTT. Equimolar amounts (20 $\mu$M) of either poly(GAAGA) or poly(AUAAU) RNA were added to poly(PR) or poly(GR) and incubated at room temperature for 30 min. Phase-separated condensates were then imaged by bright-field microscopy (M5000; EVOS). For turbidity measurements, absorbance values were read at an absorbance of 395 nm using TECAN (Safire$^2$). Statistical analyses were performed using GraphPad Prism 8. Details are given in the figure legend.

# Data Availability

The iiCLIP sequencing data are available on Gene Expression Omnibus with the accession number GSE212761.

# Supplementary Information

# Acknowledgements

The authors thank Michael Howell and the High-Throughput Screening Platform at the Francis Crick Institute for valuable assistance. R Balendra is NIHR Academic Clinical Lecturer in Neurology at UCL and has received funding from a Wellcome Trust Research Training Fellowship [107196/Z/14/Z] and the UCL Leonard Wolfson Experimental Neurology Centre for this work. She was funded by an Academy of Medical Sciences Starter Grant for Clinical Lecturers (SGL027\1022). This work was funded by the Motor Neurone Disease Association (to AM Isaacs), Alzheimer's Research UK (ARUK-PG2016A-6; ARUK-EXT2019A-002) (to AM Isaacs), the European Research Council (ERC) under the European Union's Horizon 2020 research and innovation programme (648716—C9ND) (to AM Isaacs), and the UK Dementia Research Institute (to AM Isaacs), which receives its funding from UK DRI Ltd, funded by the UK Medical Research Council, Alzheimer's Society, and Alzheimer's Research UK. HM Odeh was supported by an AstraZeneca post-doctoral fellowship and an Alzheimer's Association Research Fellowship. J Shorter was supported by ALSA, Target ALS, AFTD, and the Packard Foundation for ALS Research at JHU.

## Author Contributions

R Balendra: conceptualisation, resources, data curation, software, formal analysis, funding acquisition, validation, investigation, visualisation, methodology, project administration, and writing—original draft, review, and editing.
I Ruiz de los Mozos: resources, data curation, software, formal analysis, validation, investigation, visualisation, methodology, project administration, and writing—original draft, review, and editing.
HM Odeh: data curation, formal analysis, validation, investigation, visualisation, methodology, project administration, and writing—original draft, review, and editing.
I Glaria: investigation and writing—review and editing.
C Milioto: investigation and writing—review and editing.
KM Wilson: investigation and writing—review and editing.
AM Ule: investigation, methodology, and writing—review and editing.

M Hallegger: investigation, methodology, and writing—review and editing.

L Masino: data curation, software, formal analysis, validation, investigation, visualisation, methodology, and writing—original draft, review, and editing.

S Martin: supervision and writing—review and editing.

R Patani: supervision and writing—review and editing.

J Shorter: resources, data curation, formal analysis, supervision, funding acquisition, validation, investigation, visualisation, methodology, and writing—original draft, review, and editing.

J Ule: conceptualisation, resources, data curation, software, formal analysis, supervision, validation, investigation, visualisation, methodology, project administration, and writing—original draft, review, and editing.

AM Isaacs: conceptualisation, resources, data curation, formal analysis, supervision, funding acquisition, validation, investigation, visualisation, methodology, project administration, and writing—original draft, review, and editing.

## Conflict of Interest Statement

The authors declare that they have no conflict of interest.

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
