## [Reviewer comments · Life Science Alliance]

Life Science Alliance

Transcriptome-wide RNA binding analysis of C9orf72 poly(PR) dipeptides

Rubika Balendra, Igor Ruiz de los Mozos, Hana Odeh, Idoia Glaria, Carmelo Milioto, Katherine Wilson, Agnieszka Ule, Martina Hallegger, Laura Masino, Stephen Martin, Rickie Patani, James Shorter, Jernej Ule, and Adrian Isaacs

DOI: <https://doi.org/10.26508/lsa.202201824>

Corresponding author(s): *Adrian Isaacs, University College London*

Review Timeline:

Submission Date:	2022-11-11
Editorial Decision:	2022-12-19
Revision Received:	2023-05-23
Editorial Decision:	2023-06-20
Revision Received:	2023-06-25
Accepted:	2023-06-26

Scientific Editor: Novella Guidi

Transaction Report:

December 19, 2022

Re: Life Science Alliance manuscript #LSA-2022-01824-T

Adrian M Isaacs
UK Dementia Research Institute at UCL
Cruciform Building
90 Gower Street
London WC1E 6BT
United Kingdom

Dear Dr. Isaacs,

Thank you for submitting your manuscript entitled "Genome-wide RNA binding analysis of C9orf72 poly(PR) dipeptides" to Life Science Alliance. The manuscript was assessed by expert reviewers, whose comments are appended to this letter. We invite you to submit a revised manuscript addressing the Reviewer comments.

Thank you for this interesting contribution to Life Science Alliance. We are looking forward to receiving your revised manuscript.

Sincerely,

B. MANUSCRIPT ORGANIZATION AND FORMATTING:

Reviewer #1 (Comments to the Authors (Required)):

Summary

In this manuscript entitled: Genome-wide RNA binding analysis of C9orf72 poly(PR) dipeptides by Balendra et al. the authors provide new results showing that a polyamine, poly-PR, produced by the unconventional translation of an antisense transcript derived from the C9orf72 nucleotide repeat expansion mutation binds RNA in cells with some slight preference for specific RNA sequences, polyGAAGA. They also provide evidence that the presence of polyGAAGA, poly-PR more readily phase separates in vitro, consistent with previous studies looking at homopolymeric RNA species. This work builds on over half a century of studies examining the ability of polyamines to bind to nucleic acids and their use as methods to package or condense RNA/DNA that is in a large part driven by the strong electrostatic interactions between the negatively charged phosphate backbone of nucleic acids and the positively charged amine groups. Overall, their findings are not extraordinary, but do have some potential biological implications in C9orf72-linked neurodegeneration that could provide some utility to the field. The manuscript is also well written and has appropriate citations. However, there are significant limitations to these studies due lack of important controls that could better address specificity of these studies and improve the biological relevance of these studies, especially in the disease context. Additionally, there is limited appreciation for how this work compares or delineates from other naturally occurring polyamines, such as spermine or spermidine, and how the disease linked polyamines (PR or GR) may or may not compete with these natural substrates in physiologically relevant cellular conditions in both normal and disease settings.

Major Concerns

- 1) Representative images for PR-FLAG, GA-FLAG, and -FLAG constructs post 24s induction should be included in Figure 1 for qualitatively and quantitative assessment of expression levels and localization of the different constructs, which is important for evaluating the possible extent and likely origin of the crosslinked RNA species as depicted in figure 1c.
- 2) The use of GA as a comparison to PR is not a good control in the HEK overexpression models considering that GA often forms dense perinuclear inclusions compared to PR that is primarily nuclear. This choice should be better rationalized. It is also unclear why GR and/or GP were not examined, which would have greatly improved the robustness and the potential impact of these studies.
- 3) polyUAUAA RNA, which is predicted to form an RNA hairpin, is not a proper control for single-stranded polyGAAGA RNA - this is as much a structural comparison as a sequence comparison and therefore the authors can not state, "Interestingly, this binding shows some sequence specificity, as a higher affinity was observed with the GAAGA motif that was most enriched in the iCLIP experiment." I recommend using an alternative sequence that also has similar structures.
- 4) Extensive studies have been performed and discussed over the last half a century on the interactions of polyamines with RNA in the context of both binding preference and biological functions (for example <https://doi.org/10.1093/nar/gku837>), which overlaps with many of the findings reported in this previous studies for the polyamine poly-PR. It would be appropriate for the authors to contrast and compare such previous findings with their own current findings in the discussion and how this may be important in the context of C9orf72-linked disease.
- 5) In the discussion section, the authors should discuss how the localization of overexpressed PR may greatly influence the RNAs identified in the interactome.
- 6) In the methods section it should be mentioned if/how DNA was removed (using MNase, DNase or something similar?) to prevent DNA contamination from confounding the iCLIP PR•RNA interactions studies. If this was overlooked before generating cDNA it would be a major concern considering the predominantly nuclear localization of PR in tissue culture overexpression models and its known effects on chromatin structures
- 7) In the methods rationale should be provided for why consensus sequence analysis was limited to identifying only pentameric motifs.
- 8) It is unclear why all conditions from the iclip, PR100-crosslinked, PR100-non-crosslinked, GA100-crosslinked, GA100-non-crosslinked and FLAG-crosslinked are not included in the supplemental table, (Table S1), it seems necessary data for the readers to arrive at similar conclusions as the authors.

Minor Concerns

- 1) Authors frequently mention genome-wide (as in the title), but wouldn't it be more precise to use the term transcriptome-wide for these studies?
- 2) In figure 1b the authors should comment on the whether the levels between GA and PR are comparable in the western.

Additionally, it appears that there is well-shifting occurring in the samples that could be avoided using different denaturing protocols for badly behaving proteins.

Reviewer #2 (Comments to the Authors (Required)):

In this manuscript, Balendra et al. performed crosslinking and immunoprecipitation (CLIP) study of poly-PR, the toxic DPRs that can be RAN-translated from C9orf72 anti-sense GGCCCC RNA repeat. The authors investigated the RNA binding sites of poly(PR) in human cells, showing that nearly 600 RNAs with GAAGA sequence are enriched at the binding sites. Interestingly, they show that poly-GAAGA RNA can induce condensation of poly-PR via phase separation in vitro. Relevant to this, the authors conclude that the interaction between RNA and arginine-containing DPR can contribute to C9orf72-linked neurodegeneration.

The authors conducted a logical and well-designed study that is of sufficient interest to the Life Science Alliance readers, especially the researchers who are interested in C9orf72-ALS/FTD in association with DPR toxic gain of function. The poly-GAAGA RNA binding is new and exciting. Although there are some issues need to be resolved, this study is overall promising and the following points are recommended to be addressed before publication:

1. It is very convincing to see poly-GAAGA RNA displays higher affinity with poly-PR and induces phase separation of poly-PR into droplet-like condensates, but one may wonder whether it is linked to the disease circumstance in vivo. The authors should use culture cells to see whether poly-GAAGA RNA can affect poly-PR phase separation. Image analysis of poly-PR nucleolar localization would be an ideal assay since nucleoli are recognized as biomolecular condensates driven by protein-RNA phase separation. Given that the toxicity study is lacking in the present study, this point is crucial since nucleolar stress has been linked to arginine-containing DPR toxicity.
2. FRAP or droplet fusion analysis is needed to be performed to verify the material property of the PR condensates in Figure 5.
3. Is there any potential effect of poly-GAAGA RNA on poly-GR? The authors need to include this in the manuscript, at least in the Discussion.

Minor points:

In Figure 1B, negative control was missing. In addition, the "main band" of GA100-FLAG was missing. Was that due to GA aggregation (accumulated on the gel top)? Could the authors explain this? - similar issue in Figure S1B.

Reviewer #3 (Comments to the Authors (Required)):

Poly GA, GR, and PR are dipeptides repeat proteins, DPRs, that have been implicated in the C9orf72 hexanucleotide repeat expansion mutation linked to neurodegeneration in ALS and FTD. In this study, Balendra et al. Performed CLIP analysis in human embryonic kidney cells and provided evidence that the arginine-rich DPR, polyPR, displays enriched crosslinking to specific RNA transcripts, including ALS-relevant mRNAs such as nucleolin. The authors also showed a detailed analysis of binding affinities with GAAGA-containing RNAs, promoting phase separation of polyPR.

The authors presented a well-crafted manuscript and a potentially valuable genome-wide dataset of the polyPR interactome in human cells that could help the field understand the toxic mechanisms of the arginine-rich DPRs or the polyPR one.

Below are some comments that could be addressed experimentally or by a rebuttal.

Concerns and comments to be addressed:

Experiments are done at 24 post-transfection of the HEK cells. Is there a correlation between expression levels of polyPR and its RNA binding interactome?

Is the same 5mer motif enriched in the GA100 iiCLIP dataset?

Although this paper has been crafted to build a dataset on polyPR RNA interactome, it would be of interest to study, for disease-relevance context, whether the other arginine-rich DPR, polyGR, also binds with the same or different affinity to the same

GAAGA-containing RNAs as polyPR.

The cell line model used for the experiments limits the disease relevance of the findings.

We thank the reviewers for their insightful comments which have significantly improved our manuscript. We have addressed every comment in the point by point responses below.

Reviewer #1:

- 1) Representative images for PR-FLAG, GA-FLAG, and -FLAG constructs post 24s induction should be included in Figure 1 for qualitative and quantitative assessment of expression levels and localization of the different constructs, which is important for evaluating the possible extent and likely origin of the crosslinked RNA species as depicted in figure 1c.

We have included representative images for PR-FLAG and GA-FLAG in Figure S1 (panel A) and quantified their expression and localisation (panels B and C). We have also added the following sentence to the discussion: 'As expected, based on previous studies (Kwon et al., 2014b), PR showed some punctate nuclear staining consistent with nucleolar localisation and this is likely to greatly influence the RNAs which are found to be bound to it.' FLAG alone has a very low molecular weight and was difficult to visualise by western blotting, so we have now added a dot blot showing that the -FLAG construct is well expressed (panels D-F).

- 2) The use of GA as a comparison to PR is not a good control in the HEK overexpression models considering that GA often forms dense perinuclear inclusions compared to PR that is primarily nuclear. This choice should be better rationalized. It is also unclear why GR and/or GP were not examined, which would have greatly improved the robustness and the potential impact of these studies.

We thank the reviewer for this comment. We specifically used PR100-non-crosslinked and FLAG-crosslinked as controls for the iiCLIP experiments, as we were interested in the specific binding of poly(PR) to RNA. We used GA100 as a further condition. We have added the sentence 'As controls to test for specific binding of poly(PR) to RNA, we used poly(PR) expressing cells which were non-crosslinked, and FLAG expressing cells which were crosslinked' to further rationalise the controls used.

We had planned to use GR for iiCLIP as well, but the expression was not good enough and combined with its aggregation we were never happy enough with its expression to go forward. Therefore, although we did not use poly(GP) or poly(GR) in the iiCLIP experiments, we have used poly(GP) in the affinity experiments (Figure 4), showing that this does not interact with RNA. We have also now examined poly(GR) in the affinity and phase separation experiments (Figures 4 and 5), to investigate its interaction with the enriched

RNA sequence found. We have found that poly(GR) also has a high affinity for poly(GAAGA) RNA, and this is slightly higher than its affinity for the control poly(AUAAU) RNA. We have now also included poly(GR) in the condensation assay. Similarly to poly(PR), we found that poly(GAAGA) induces poly(GR) condensation and that the phase separation of poly(GR) is markedly reduced with poly(AUAAU).

- 3) polyUAUAA RNA, which is predicted to form an RNA hairpin, is not a proper control for single-stranded polyGAAGA RNA - this is as much a structural comparison as a sequence comparison and therefore the authors can not state, "Interestingly, this binding shows some sequence specificity, as a higher affinity was observed with the GAAGA motif that was most enriched in the iCLIP experiment." I recommend using an alternative sequence that also has similar structures.

We have now used a control RNA sequence which does not form a hairpin (poly(AUAAU)), and we have updated the experiments in Figure 4 and 5 with this, demonstrating a higher affinity between poly(PR) and poly(GAAGA) RNA compared to poly(AUAAU) RNA. We have also shown that the phase separation of poly(PR) and poly(GR) is markedly reduced with the poly(AUAAU) RNA compared to poly(GAAGA) RNA.

Extensive studies have been performed and discussed over the last half a century on the interactions of polyamines with RNA in the context of both binding preference and biological functions (for example <https://doi.org/10.1093/nar/gku837>), which overlaps with many of the findings reported in this previous studies for the polyamine poly-PR. It would be appropriate for the authors to contrast and compare such previous findings with their own current findings in the discussion and how this may be important in the context of C9orf72-linked disease.

We thank the reviewer for this important literature. We believe it is beyond the scope of this manuscript to discuss the polyamine literature, but could be the topic of a future review.

- 4) In the discussion section, the authors should discuss how the localization of overexpressed PR may greatly influence the RNAs identified in the interactome.

We have added the following sentence to the discussion: 'As expected, based on previous studies (Kwon et al., 2014b), PR showed some punctate nuclear staining consistent with nucleolar localisation and this is likely to greatly influence the RNAs which are found to be bound to it.'

- 5) In the methods section it should be mentioned if/how DNA was removed (using MNase, DNase or something similar?) to prevent DNA contamination from confounding the iCLIP PR•RNA interactions studies. If this was overlooked before generating cDNA it would be a major concern considering the predominantly nuclear localization of PR in tissue culture overexpression models and its known effects on chromatin structures

We thank the reviewer for pointing out this important point. DNase was used after cell lysis to remove DNA – we have now added this to the methods.

- 6) In the methods rationale should be provided for why consensus sequence analysis was limited to identifying only pentameric motifs.

We have added the following sentence to the methods: ‘Pentamers were chosen as they have proven in previous systematic studies most useful to distinguish the sequence binding specificity of RBPs both for analysis of *in vitro* binding specificity from methods such as RNA Bind-n-Seq (Dominguez et al., 2018) and for analysis of CLIP data (Kuret et al., 2022), and thus the results of our analyses can be most easily compared to other studies.’

8 It is unclear why all conditions from the iclip, PR100-crosslinked, PR100-non-crosslinked, GA100-crosslinked, GA100-non-crosslinked and FLAG-crosslinked are not included in the supplemental table, (Table S1), it seems necessary data for the readers to arrive at similar conclusions as the authors.

We have now included all the iiCLIP conditions in Table S1.

Minor Concerns

- 1) Authors frequently mention genome-wide (as in the title), but wouldn't it be more precise to use the term transcriptome-wide for these studies?

We have changed the term to transcriptome-wide throughout.

- 2) In figure 1b the authors should comment on the whether the levels between GA and PR are comparable in the western. Additionally, it appears that there is well-shifting occurring in the samples that could be avoided using different denaturing protocols for badly behaving proteins.

We have quantified expression of PR and GA in the westerns and shown they are comparable, we now include this data as Figure 1C. We have previously tried a denaturing protocol with urea cell lysis, and observed the same pattern of migration of these proteins due to their aggregation propensity so have not been able to improve the blot beyond what is shown.

Reviewer #2:

The authors conducted a logical and well-designed study that is of sufficient interest to the Life Science Alliance readers, especially the researchers who are interested in C9orf72-ALS/FTD in association with DPR toxic gain of function. The poly-GAAGA RNA binding is new and exciting. Although there are some issues need to be resolved, this study is overall promising and the following points are recommended to be addressed before publication:

1. It is very convincing to see poly-GAAGA RNA displays higher affinity with poly-PR and induces phase separation of poly-PR into droplet-like condensates, but one may wonder whether it is linked to the disease circumstance in vivo. The authors should use culture cells to see whether poly-GAAGA RNA can affect poly-PR phase separation. Image analysis of poly-PR nucleolar localization would be an ideal assay since nucleoli are recognized as biomolecular condensates driven by protein-RNA phase separation. Given that the toxicity study is lacking in the present study, this point is crucial since nucleolar stress has been linked to arginine-containing DPR toxicity.

We thank the reviewer for this thoughtful comment and agree that this would be a very insightful experiment. However, given the complexities of such an experiment in terms of RNA delivery, detection and imaging it is beyond the scope of the current study. We do agree this will be important to perform in follow up work and have now added this suggestion to the discussion with the sentence: 'It would be of interest in future studies to determine whether these RNA-induced condensates are less toxic to cells and whether poly(GAAGA) is able to alter phase separation of poly(PR) within cells.'

2. FRAP or droplet fusion analysis is needed to be performed to verify the material property of the PR condensates in Figure 5.

We agree that it would be interesting to determine the material properties of the condensates, but determining whether the condensates are liquid or solid is beyond the scope of the current study, which is mostly focused on our iiCLIP data. However, in order to acknowledge your point and to more accurately describe the condensates we have dropped "liquid-liquid" from "liquid-liquid phase separation" throughout the paper, and now simply refer to it as phase separation.

3. Is there any potential effect of poly-GAAGA RNA on poly-GR? The authors need to include this in the manuscript, at least in the Discussion.

We thank the review for raising this, which we agree is important to address. We have now examined poly(GR) in the affinity and phase separation experiments, to investigate its interaction with the enriched RNA sequence found. We have found that poly(GR) also has a high affinity for poly(GAAGA) RNA, and this is slightly higher than its affinity for the control poly(AUAAU) RNA. Interestingly poly(PR) showed a greater differential affinity, which is consistent with our PR100 iiCLIP data. We have now also included poly(GR) in the condensation assay. Similarly to poly(PR), we found that poly(GAAGA) induces poly(GR) condensation and that the phase separation of poly(GR) is markedly reduced with poly(AUAAU).

Minor points:

In Figure 1B, negative control was missing. In addition, the "main band" of GA100-FLAG was missing. Was that due to GA aggregation (accumulated on the gel top)? Could the authors explain this? - similar issue in Figure S1B.

We have now included a FLAG only negative control in Figure 1B. Unfortunately, the FLAG protein has a very low molecular weight and despite several efforts we have not been able to visualize it by western blot. However, we have still included the gel in Figure 1B to be as clear as possible. We have now added a dot blot showing that FLAG is well expressed (Figure S1, panels D-F). In our hands GA100 is highly aggregation prone and tends to aggregate at the top of the gel, rather than producing a visible band at the expected molecular weight. We have now explained this in the figure legend 'Note GA100 appears mostly aggregated as the majority is present at the top of the gel'.

Reviewer #3:

The authors presented a well-crafted manuscript and a potentially valuable genome-wide dataset of the polyPR interactome in human cells that could help the field understand the toxic mechanisms of the arginine-rich DPRs or the polyPR one.

Below are some comments that could be addressed experimentally or by a rebuttal.

Concerns and comments to be addressed:

Experiments are done at 24 post-transfection of the HEK cells. Is there a correlation between expression levels of polyPR and its RNA binding interactome?

We appreciate this thoughtful point. It is not clear whether absolute levels of PR would affect its RNA binding interactome, but it is something to consider when using an over-expression paradigm such as this one. We have now added some text to the discussion, to highlight the need to further confirm results in the endogenous setting, ideally in neurons.

Is the same 5mer motif enriched in the GA100 iiCLIP dataset?

We have analysed the most enriched 5mer in the GA100 iiCLIP dataset which is CCGGG and is different to the PR100 iiCLIP dataset — we have now added this analysis as supplemental figure 4.

Although this paper has been crafted to build a dataset on polyPR RNA interactome, it would be of interest to study, for disease-relevance context, whether the other arginine-rich DPR, polyGR, also binds with the same or different affinity to the same GAAGA-containing RNAs as polyPR.

Based on this comment we have also now examined poly(GR) in the affinity experiments, to investigate its interaction with the enriched RNA sequence found. We have found that poly(GR) also has a very high affinity for poly(GAAGA) RNA, and this is slightly higher than its affinity for the control poly(AUAAU) RNA. Interestingly poly(PR) showed a greater differential affinity, which is consistent with our PR100 iiCLIP data.

The cell line model used for the experiments limits the disease relevance of the findings.

We have added the following sentence to the discussion: 'This study has been performed in human cell lines using over-expression of poly(PR), which limits the disease relevance of these findings, and future studies in neuronal models with more physiological expression levels would be of importance.'

June 20, 2023

RE: Life Science Alliance Manuscript #LSA-2022-01824-TR

Prof. Adrian M Isaacs
University College London
Neurodegenerative Disease
Queen Square
London WC1N 3BG
United Kingdom

Dear Dr. Isaacs,

Thank you for submitting your revised manuscript entitled "Transcriptome-wide RNA binding analysis of C9orf72 poly(PR) dipeptides". We would be happy to publish your paper in Life Science Alliance pending final revisions necessary to meet our formatting guidelines.

- please add an Author Contributions section to your main manuscript text
- please add a conflict of interest statement to your main manuscript text
- please use the [10 author names et al.] format in your references (i.e., limit the author names to the first 10)
- please add your main, supplementary figure, and table legends to the main manuscript text after the references section
- it is recommended to exclude figures from the manuscript text and upload them separately
- please add a callout for Figure 1D to your main manuscript text

Figure checks:

- please improve the quality of the blots in Figure 1B.
- In Figure 2A the contrast seems to be overexposed. Please provide a representative blot image not overexposed.

A. FINAL FILES:

B. MANUSCRIPT ORGANIZATION AND FORMATTING:

Sincerely,

Reviewer #1 (Comments to the Authors (Required)):

The authors have adequately addressed all of my previous comments in this round of revision, and I find it now appropriate for publication in Life Science Alliance.

Reviewer #2 (Comments to the Authors (Required)):

The authors have addressed the previous concerns, and the manuscript looks much more solid.

Reviewer #3 (Comments to the Authors (Required)):

The authors have adequately addressed the concerns raised.

I believe the manuscript is now suitable for publication.

The revised manuscript has effectively incorporated the feedback provided by the reviewers, resulting in a significant improvement in its quality and clarity. The authors have diligently addressed all the major concerns and have made appropriate changes throughout the manuscript. The revisions have significantly strengthened the overall structure, methodology, and findings of the study, making it a valuable contribution to the field.

Strengths:

Robust data analysis: The revised data analysis now provides a more comprehensive understanding of the research question, leading to more reliable conclusions.

Expansion of discussion: The revised manuscript includes an expanded discussion section that incorporates insights from the additional analyses and addresses the limitations of the study. The authors have also included relevant references suggested by the reviewers, which enhances the scientific context and ensures a comprehensive literature review.

The study is significant, the methodology is sound, and the findings contribute to the existing body of knowledge in the field.

June 26, 2023

RE: Life Science Alliance Manuscript #LSA-2022-01824-TRR

Prof. Adrian M Isaacs
University College London
Neurodegenerative Disease
Queen Square
London WC1N 3BG
United Kingdom

Dear Dr. Isaacs,

Thank you for submitting your Research Article entitled "Transcriptome-wide RNA binding analysis of C9orf72 poly(PR) dipeptides". It is a pleasure to let you know that your manuscript is now accepted for publication in Life Science Alliance. Congratulations on this interesting work.

DISTRIBUTION OF MATERIALS:

Again, congratulations on a very nice paper. I hope you found the review process to be constructive and are pleased with how the manuscript was handled editorially. We look forward to future exciting submissions from your lab.

Sincerely,
